# Mimicking the Fungal Decay Strategy for Promoting the Bacterial Production of Polyhydroxyalkanoate from Kraft Lignin

**Xiao Fu [†], Qing Gong [†], Xuan Liu, Ze Zheng, Xiaoyu Zhang, Fuying Ma [ID], Hongbo Yu *[ID] and Shangxian Xie ***

Key Laboratory of Molecular Biophysics of MOE, College of Life Science and Technology, Huazhong University of Science and Technology, Wuhan 430074, China; fuxiao@hust.edu.cn (X.F.)
* Correspondence: yuhongbo@hust.edu.cn (H.Y.); shangxian_xie@hust.edu.cn (S.X.)
† These authors contributed equally to this work.

**Abstract:** Producing polyhydroxyalkanoate (PHA) from lignin through biological conversion has great further potential, but is held by its high heterogeneous characteristic toxicity of the depolymerized products, and low bioconversion of the depolymerized products. In this study, a Fenton-like reaction, which is inspired by fungal decay strategy, was reported to cleave Kraft lignin linkages and produce low toxic mono-aromatic and low molecular organic compounds for microbial conversion. The treatment improved the bioconversion of lignin to PHA by 141.7% compared to Kraft lignin. The bond cleavage of Kraft lignin was characterized by Py-GC/MS and 2D NMR. Seven major depolymerized products were chosen to test their toxicity effect on bacterial fermentation. Furthermore, 920.4 mg of PHA was obtained from 1-L black liquor after Fenton-like reaction treatment. This is a novel attempt mimicking fungal decay strategy coupled with the microbial conversion of lignin into high-value PHA with a sustainable future.

**Keywords:** Kraft lignin; Fenton-like reaction; depolymerization; PHA; lignin

## 1. Introduction

Petroleum shortage and environmental concerns have made the biofuel from lignocellulose represent the backbone of biorefinery. Translationally, lignocellulose biorefinery and the pulping industry usually focus on the conversion of cellulose and hemicellulose into energy and high-value chemicals, while leaving the high-energy density aromatic biopolymer, lignin, as an underutilized waste for heat and power production [1,2]. The annual market value of lignin, comprising 15–40% dry weight of lignocellulose, which is the second most abundant biopolymer after cellulose in this planet, reached 732.7 million US dollars in 2015, with an expected global market size up to $913.1 million by 2025 [3]. Kraft lignin from black liquor of the Kraft pulping process constitutes the largest technical lignin stream by volume, and its quantity is estimated to be over 630,000 ton annually [4]. Recently, many researchers have recognized the potential for the conversion of Kraft lignin into commercial products such as liquid transportation fuel and value-added chemicals [5]. However, Kraft lignin has more abundance of stable C-C linkages than other lignin because of the extensive chemical transformation of native lignin in the high-severity pulping process [6]. The undesirable, highly complex, and condensed structure of Kraft lignin renders it challenging for feedstocks for lignin valorization.

Ligninolytic bacteria, such as *Pseudomonas putida* and *Cupriavidus basilensis*, can directly convert lignin-derived aromatic compounds into polyhydroxyalkanoates (PHA), which has been advanced for lignin valorization [7–9]. PHA have been widely used in bioplastic and biomedicine due to their excellent biodegradability and biocompatibility [10]. Although these bacteria conversion methods have some advantages, including their strong environmental adaptability and rapid growth, efficient bacterial conversion of recalcitrant and heterogeneous Kraft lignin to lignin-derived aromatic compounds remains a challenge [11]. Previous reports showed that the yield of PHA from Kraft lignin was only about

30–120 mg/L [9,12], due to their inefficient lignin depolymerization capacity. Therefore, a highly efficient strategy to degrade lignin into low molecular products for improving its microbial conversion represents the current need for lignin valorization.

Various chemical, thermochemical, and biological routes have been used in Kraft lignin valorization. Among those routes, the biological process has received more attention due to its environmentally benign, less energy consumption, and less toxicity compounds production [13]. In nature, a number of microorganisms, including fungi and bacteria, are able to depolymerize lignin [14]. Wood-decay fungi, which is a critical decomposer of terrestrial plant biomass, can effectively depolymerize and mineralize lignin using not only ligninolytic enzymes, such as laccase and peroxidase, but also radical-mediated oxidative reactions [15]. These fungi can generate hydroxyl radical from extracellular $H_2O_2$ and chelated iron complexes that enhance the solubilization and reduction of iron though a Fenton-like reaction for an efficient depolymerization of lignin [16]. Compared to the Fenton reaction, the Fenton-like reaction has higher hydroxyl radical yield and its initial pH values are similar to the natural environment of decay. Some studies have reported that wood-decay fungi can depolymerize the lignin to lignin-derived aromatic compounds from black liquor [17]. However, few species of wood-decay fungi can use lignin-derived aromatic compounds as the sole carbon source because sugar is prerequisite to their growth, which limits the potential of fungal conversion in lignin valorization. As a result, a highly efficient method to depolymerize lignin at natural conditions and generate less toxicity products and further accelerate subsequent bacterial conversion process is promising for lignin valorization.

In this study, we mimicked the fungal decay strategy to develop an efficient method to accelerate hydroxyl radical production by using $H_2O_2$ and the chelated iron complex Fe-ethylenediaminetetraacetic acid (EDTA), which can enhance the process of depolymerized lignin to lignin-derived mono-aromatic and short-chain organic compounds. This Fenton-like reaction system was used to improve subsequent bacterial conversion of treated Kraft lignin into PHA. This is the first time that the strong lignin-oxidizing capacity of fungus has been combined with bacterial fermentation for converting Kraft lignin to PHA. The structural changes and lignin breakdown products during lignin depolymerization were also investigated by pyrolysis–gas chromatography/mass spectrometry (Py–GC/MS), two-dimensional nuclear magnetic resonance (2D NMR) spectroscopy, and gas chromatography-mass spectrometer (GC-MS). Black liquor from the Kraft pulping industry was used to further test the potential of this oxidative reaction for practical applications. By developing the radical depolymerization and bioconversion process of lignin for PHA production, a nature-inspired, sustainable, and highly efficient method is demonstrated to be suited for lignin valorization.

## 2. Materials and Methods

### 2.1. Pretreatment of Kraft Lignin by Fenton-like Reaction

$FeSO_4$ and EDTA disodium salt were diluted to the desired concentrations and were mixed at molar ratio 2:1. Kraft lignin (Sigma-Aldrich, Saint Louis, MO, USA) was suspended in a $FeSO_4$-EDTA solution at a solid loading of 5% (*w/v*), $H_2O_2$ (30% *w/w*) was added, and the reaction was carried at 55 °C with shaking at 150 rpm for 24 h, as reported by previous work [18]. To study the effect of reaction system on radical generation and lignin polymerization, different concentrations of reagent were used (see the Supplementary Materials Table S1). After incubating for desired pretreatment durations, the liquid was heated to 120 °C for 30 min. After cooling to room temperature, the liquid was adjusted to pH 5 with 10 M NaOH and filtered through Whatman no. 54 filter paper. The culture was heated to 120 °C for 20 min, and after cooling to room temperature, the liquid was adjusted to pH 7 with 10 M NaOH and filtered through Whatman no. 54 filter paper. The filtrated liquid should be stored for at least 12 h before bacterial cultivation. All other chemicals and reagents were purchased from Sinopharm Chemical Reagent Co., Ltd. (Shanghai, China).

### 2.2. Analysis of Hydroxyl Radical and Lignin Concentration

A deoxyribose assay was used to detect the concentration of hydroxyl radicals [19], and the Fenton reaction samples were taken from the reaction system at different time points (5 s, 30 min, 1 h, 2 h, 4 h, 9 h, and 24 h). Immediately after sampling, 10 µL of sample was added to 600 µL of 0.1 M 2-deoxyribose, 900 µL of 50 mM pH 7.0 sodium phosphate buffer, and 490 µL of $H_2O$ in Eppendorf tube, and incubated at 37 °C for 2 h. Then, 200 µL of reaction mixture was taken out and added with 0.2 mL of 2.8% trichloroacetic acid and 0.2 mL of 1.0% 2-tribarbitric acid in 50 mM NaOH, and the solution was boiled for 10 min to stop the reaction. After cooling in water, the absorbance of the solution was measured at 530 nm to determine the intensity of the free radical. The lignin in solution (initial concentration 50 g/L) was precipitated by 1 mL of 0.6 M HCl and separated by centrifuge (12,000 rpm, 5 min), and the residue was dissolved in 1 mL $ddH_2O$ for the determination of lignin concentration by the absorbance at 278 nm.

### 2.3. Bacterial Strains and Cultivation

The bacteria used in this study was *Pseudomonas putida* KT2440, obtained from Nanjing Agricultural University (Nanjing, Jiangsu, China). Cells were cultured in 20 mL Luria-Bertani (LB) broth medium (10 g/L peptone, 5 g/L yeast extract, and 0.5 g/L sodium chloride) overnight at 30 °C with shaking at 200 rpm. The 20 mL pre-culture strain was inoculated into 100 mL fresh LB medium at 28 °C with shaking at 200 rpm to an optical density at 600 nm of inoculums reaching 1.6. Bacterial cells were washed three times with control medium (100 mL control medium contains 88 mL $ddH_2O$, 10 mL $10\times$ M9 minimal medium, 1 mL $100\times$ Mg/Ca/B1/Goodies mix; details of the medium formula are shown in the Supporting Information), centrifuged (4 °C, 6000 rpm, 5 min), and resuspended with 6 mL control medium for further inoculation. Then, 1 mL of concentrated bacteria cells was inoculated into 99 mL of growth medium (88 mL diluted lignin or lignin-derived aromatic compounds solution, 10 mL $10\times$ Basal salts, 1 mL $100\times$ Mg/Ca/B1/Goodies mix) and cultivated at 28 °C with shaking at 200 rpm.

### 2.4. Extraction of PHA Form Cells

After a certain time of cultivation in a different medium, bacterial cells were harvested by centrifugation at 10,000 rpm for 10 min at 4 °C, washed twice with deionized water, lyophilized for 24 h, and weighted to get the dry cell mass. The obtained bacterial cells were suspended in 7 mL chloroform and incubated for 12 h at 60 °C with shaking at 150 rpm to extract the PHA [20]. Then, 2 mL of $ddH_2O$ was added to the organic mixture. The organic phase was collected and filtered through 0.45 µm polytetrafluorethylene (PTFE) membrane, and then concentrated to 1 mL by $N_2$ flux. The PHA was isolated and purified by precipitation through the addition of 10 mL pre-chilled methanol and allowed to settle down for 30 min. The white precipitate formed was filtered, air-dried, and weighted. The yield of PHA was calculated using the following equations:

$$\text{PHA yield (mg/L)} = \frac{W_{PHA}}{V_m} \tag{1}$$

$$\text{PHA content (\%)} = \frac{W_{PHA}}{W_B} \tag{2}$$

where $W_{PHA}$ is the weight of PHA, $V_m$ is the volume of medium, and $W_B$ is the weight of bacterial biomass used for PHA extraction.

### 2.5. Kraft Lignin Characterizations

Py-GC/MS analysis was used to determine the chemical structure changes of Kraft lignin by the Fenton-like reaction. A volume of 10 mL of Kraft lignin solution and Fenton-like-reaction-treated samples were lyophilized. Pyrolysis was carried out with a Pyroprobe 5200 analytical pyrolyzer (CDS Analytical, Oxford, PA, USA) coupled to a GC 7890A

5975CMSD gas chromatography/mass spectrometry system (Agilent Technologies, Aald-bronn, Germany) to analyze the volatiles. Pyrolysis products were assigned based on the NIST mass spectrum library and our previous method [21].

The structures of the Kraft lignin and Fenton-like-reaction-treated lignin were analyzed by using an Agilent DD2 600 MHz superconducting NMR spectrometer with the "gHSQCAD" pulse sequence. The isolated lignin (100 mg) samples were resuspended in 0.75 mL of dimethylsulfoxide (DMSO)-*d6* in the NMR tube. HSQC experiments were carried out at 35 °C on an Agilent DD2 600 MHz superconducting NMR spectrometer with the "gHSQCAD" pulse sequence based on previously published work [22].

The procedure for determining the methoxyl (OMe) content of lignin samples was modified from the literature [23]. Briefly, 20 mg of lignin was added to 3 mL of 57% (*w/v*) hydriodic acid in a 50 mL head-space vial, which was sealed with teflon/silicone septa and a crimped cap. The vial was filled with $N_2$ gas and reacted in 140 °C welled hot-plat for 90 min. After the reaction, the vial was cooled down in an ice-water bath immediately, followed by adding 6 mL of $CHCl_3$, and then the vial was shaken vigorously for 1 min. The vial was placed in −20 °C for 1 h to allow complete phase separation. After separation, 1 mL of mixture was filtrated with a microfilter (0.22 μm) to remove the particles in the organic phase. The OMe content was quantitated by the GC/MS external standard method as reported previously [23], and calculated by the following equations:

$$\text{OMe (mg)} = (\text{Ave mg/mL CH3I}) \times (6 \text{ mL total volume}) \times (1 \text{ mmol CH3I}/141.95 \text{ mg CH3I}) \times (1 \text{ mmol OMe}/1 \text{ mmol CH3I}) \times (31.03 \text{ mg OMe}/1 \text{ mmol OMe}) \tag{3}$$

$$\text{OMe (\%)} = \frac{M_{OMe}}{M_S} \times 100\% \tag{4}$$

where $M_{OMe}$ and $M_S$ are the masses of OMe and the dry sample, respectively.

### 2.6. Detection of Lignin Depolymerized Products

The products from lignin depolymerization were analyzed by GC-MS. Kraft lignin solution, Fenton-like-reaction-treated lignin solution, and bacterium-inoculated treated lignin supernatant (centrifuged at 10,000 rpm for 10 min to remove bacterial strains) were adjusted to pH 1–2 with concentrated HCl. The supernatants were extracted with three volumes of ethyl acetate. The organic layer was collected, dewatered over anhydrous $Na_2SO_4$, and filtered through Whatman no. 54 filter paper. The residues were concentrated by rotary evaporation and then dried under a stream of nitrogen gas. A volume of 500 μL dioxane and 100 μL pyridine were added to the dried residues and silylated with 300 μL BSTFA-TMCS (99:1). The mixture was heated at 80 °C for 20 min [11]. After reaction, the mixture was filtrated and 1 uL of the sample was injected into GC/MS for analysis. The column temperature program was 50 °C (5 min), 50–300 °C (10 °C/min, hold time: 15 min). The transfer line and ion source temperatures were maintained at 200 °C and 250 °C, respectively. A solvent time delay of 7.5 min was selected [24]. The details of the detection of the concentration of glucose and lignin depolymerized products are provided in the Supporting Information.

### 2.7. Black Liquor Fermentation

Black liquor used in this study was obtained from Yueyang Paper Industry Co., Ltd. (Yueyang, China). The black liquor was adjusted to pH 7.0 by 1 M HCl and precipitated at 28 °C overnight. To study the potential of soluble compounds in black liquor as the solo carbon source for PHA accumulation, the liquid phase was diluted five times for bacterial fermentation. Additionally, the collected liquid phase was subjected to GC/MS for analyzing the composition of soluble compounds; meanwhile, the structure of soluble lignin was analyzed through 2D NMR. The solid residue which had similar high recalcitrant and toxic characteristics as the Kraft lignin used above, and, therefore, was treated with a Fenton-like reaction. After the treatment, the solid lignin was depolymerized into small molecular compounds and further dissolved in the liquid phase. The treated solid residue

was used for fermentation as mentioned above, and the PHA accumulation after the bacterial conversion was detected.

## 3. Results

### 3.1. Fenton-like Reaction Treatment

The Fenton-like reaction system can produce a high concentration of hydroxyl radical (Figure 1A,B), and the concentration of radical increased with increasing concentrations of chelated iron and hydrogen peroxide used in the reaction. However, the radical was quenched with the reaction time extended. As shown in Figure 1A, after 4 h of reaction, the radical concentration was 57.0, 38.8, and 15.1% lower than it was at the beginning of the reaction on 2 M $H_2O_2$ reacting with 1 mM, 2 mM, and 4 mM $Fe^{2+}$, respectively. For the 4 M $H_2O_2$ addition system, the corresponding decrease was 84.1, 61.1, and 35.3%, respectively (Figure 1B). This result suggests that the radical generated in the Fenton-like reaction system is unstable. Interestingly, the concentration of radical increased, along with the reaction time, after lignin was added to the reaction, which is significantly higher than without lignin addition during the 24 h of reaction (Figure 1A,B). As reported by previous research, the radical was stabled by the products derived from lignin degradation such as organic acids and aromatic compounds [25]. The radical production was stable and enhanced by lignin and lignin degradation products. These results show the potential of lignin assisted the Fenton-like reaction efficiently and stably to produce a high yield of free radicals.

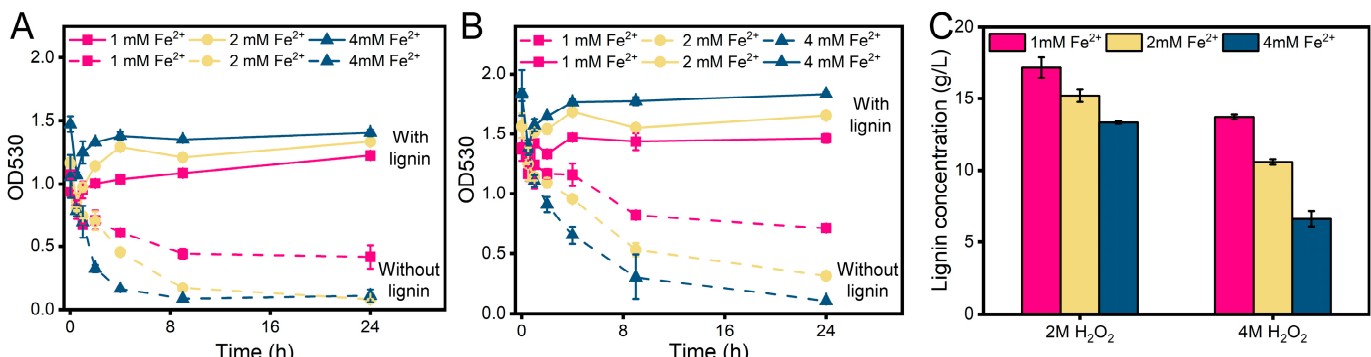

**Figure 1.** The effect of Fenton-like reaction system on the free radical production and lignin degradation. Free radical production of Fenton-like reaction system with (**A**) 2 M $H_2O_2$ and (**B**) 4 M $H_2O_2$. (**C**) Lignin concentrations after Fenton-like reaction system treated for 24 h.

The lignin degradation results proved the capacity of the Fenton-like reaction to depolymerize lignin. As shown in Figure 1C, the lignin concentration was decreased after 24 h of treatment with the Fenton-like reaction, and the concentration of lignin decreased from 50.0 to $17.2 \pm 0.5$, $15.2 \pm 0.3$, and $13.4 \pm 0.2$ g/L after treatment of 2 M $H_2O_2$ reacted with 1 mM, 2 mM, and 4 mM Fe2+, respectively. The corresponding concentration was $13.7 \pm 0.3$, $10.6 \pm 0.3$, and $6.6 \pm 0.5$ after treated with 4 M $H_2O_2$, respectively. The lignin degradation was enhanced by the improvement of radical concentration; the maximum lignin degradation was 86.7%, which was obtained at 4 M $H_2O_2$ reacting with 4 mM $Fe^{2+}$ (Figure 1C). These results highlight the capacity of the Fenton-like reaction for lignin degradation via stable radical production.

### 3.2. PHA Production by P. putida KT2440 Fermentation

*P. putida* KT2440 was used for PHA accumulation on Kraft lignin and treated lignin. Although *P. putida* KT2440 can produce PHA using 0.5% (*w/v*) Kraft lignin as the sole carbon source, the maximum PHA yield and bacterial biomass were only $63.1 \pm 3.5$ and $320.5 \pm 40.2$ mg/L (Figure 2A,B), respectively, due to the high recalcitrance of lignin for bacterial fermentation [2]. The PHA yield decreased when the concentration of Kraft lignin

further increased (Figure 2A), and when the lignin concentration increased to 3.75%, cell growth and PHA production were completely inhibited (Figure 2A,B), probably due to the toxic lignin-derived products generated from the chemical fractionation method (Kraft pulping) that inhibit microbial growth [13]. Based on this, we employed the Fenton-like reaction, which is a biomimetic reaction of the fungal delignification strategy, to depolymerize kraft lignin into low molecular compounds which could be used for microbial growth.

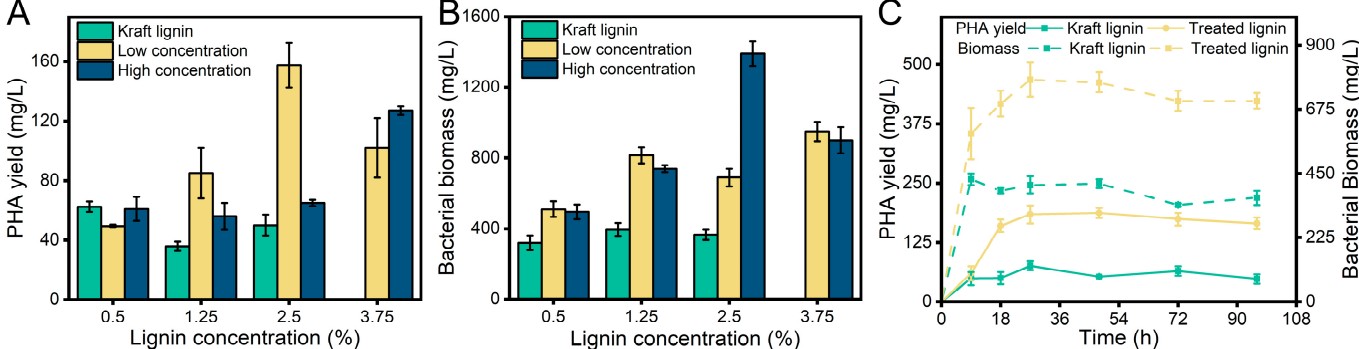

**Figure 2.** Biomass and PHA yield of *P. putida* KT2440 on lignin. (**A**) PHA yield and (**B**) bacterial biomass of *P. putida* KT2440 with different lignin concentration ranges from 0.5–3.75%. (**C**) PHA yield and bacterial biomass of *P. putida* KT2440 with 2.5% of Kraft and treated (2 M $H_2O_2$ and 2 mM $Fe^{2+}$) lignin addition.

Lignin degradation was enhanced by the high concentration of iron and hydrogen peroxide during the reaction, as mentioned above (Figure 1C). To investigate the effect of Kraft lignin degradation on bacterial growth and PHA production, two groups with high (4 mM $Fe^{2+}$, 4 M $H_2O_2$) and low (2 mM $Fe^{2+}$, 2 M $H_2O_2$) reaction reagents were chosen for Kraft lignin depolymerization. As shown in Figure 2A,B, the PHA yield and bacterial biomass were $49.5 \pm 1.0$ and $510.2 \pm 42.5$, $85.2 \pm 815.3$ and $157.5 \pm 690.3$, $102.3 \pm 20.1$ mg/L and $950.2 \pm 55.8$ mg/L with 0.5, 1.25, 2.5, and 3.75% of treated lignin (low concentration) addition, respectively, which were 1.6–3.2-fold of Kraft lignin with the corresponding concentration. For the lignin treated with the high concentration, the bacterial biomass and PHA yield were also improved greatly (Figure 2A,B). Lignin treated with a high iron concentration enhanced bacterial growth, whereas lignin treated with a low iron concentration was more pronounced for PHA production. The highest PHA yield (157.5 mg/L) was obtained by using 2.5% of low concentration treated lignin as a solo carbon source, which increased by 152% compared to Kraft lignin. As a result, 2 mM $Fe^{2+}$ and 2 M $H_2O_2$ (low concentration) were the optimized conditions of the Fenton-like reaction for lignin depolymerization, and 2.5% of treated lignin was chosen for enhanced PHA production from *P. putida* KT2440 fermentation.

The production of PHA and the bacterial biomass of *P. putida* KT2440 increased with time from 0 to 27 h, and were found to be $76.2 \pm 8.9$ mg/L and $410.2 \pm 28.7$ mg/L on Kraft lignin and $184.2 \pm 16.7$ mg/L and $780.5 \pm 60.3$ mg/L on treated lignin (Figure 2C), respectively. The corresponding PHA content was found to be $18.5 \pm 0.3$% and $23.6 \pm 0.5$%, respectively. The accumulation of PHA from the treated lignin is 2.4-fold from the Kraft lignin. There was a slight decrease in the PHA accumulation and bacterial biomass measured after 27 h (Figure 2C). These results indicate that the Fenton-like reaction system can significantly improve the efficiency of microbial conversion on Kraft lignin. This enhancement might be attributed to the increased low-toxic lignin-derived compounds and decreased recalcitrant of lignin. Thus, the structure changes of lignin and the mechanism of the Fenton-like-reaction-assisted lignin microbial conversion were then elucidated by Py-GM/MS, 2D NMR, GC/MS, and lignin-derived compounds fermentation.

### 3.3. Characterization of Kraft Lignin

Lignin structure and lignin-derived aromatic compounds changes revealed that the Fenton-like reaction degraded lignin through carbon-oxygen (C-O) and carbon-carbon (C-C) bonds cleavage at natural pH and at ambient pressure. The Kraft lignin was mainly depolymerized into mono-aromatic and short chain organic acid compounds, which further enhanced PHA production through microbial conversion.

First, approximately 30 phenolic compounds were identified from Py-GC/MS. As shown in Table 1, the guaiacyl (G-type) lignin derivatives were predominant products, and a few ρ-hydroxyphenyl (H-type) lignin and zero sinapyle-type lignin indicated that the Kraft lignin used in this study was isolated from black liquor which was pulped with coniferous wood [26]. After the Fenton-like reaction treatment, the total abundance of G-type lignin derivatives significantly decreased from 80.02% to 56.44%. Furthermore, 2-methoxy-5-methylphenol, 2-methoxy-4-vinylphenol, vanillin and 1-(4-hydroxy-3-methoxyphenyl)-ethanone in treated lignin could not be detected (Table S2). The abundance of H-type lignin derivatives for treated lignin was found to be 43.56%, which was 6.71-fold of that in Kraft lignin. As a result, the G/H ratio of lignin significantly decreased from 12.33 to 1.30 after the Fenton-like reaction treatment (Table 1). This result confirms the previous observation showing the decrease of the G/H ratio in fungal or Fenton-reagent-treated biomass [27,28]. This phenomenon is mainly caused by the demethylation of G-type lignin during the treatment [27]. Compared to H-type lignin, the G-type lignin includes a high degree of methoxylation and a lower percentage of C-C, leading to a higher predominance of β-O-4 linkages. The β-O-4 linkages were less recalcitrant for radical attack [29]. The decreased G/H ratio in treated lignin suggests that the Fenton-like reaction treatment selectively cleaved β-O-4 linkages in lignin and led to the reduced recalcitrate of lignin for microbial growth and PHA production (Figure 2C).

**Table 1.** Py-GC/MS relative abundance of Kraft lignin and Fenton-like-reaction-treated lignin.

|  | **Kraft Lignin** | **Treated Lignin** |
|---|---|---|
| Lignin subunits (%) | | |
| H | 6.49 | 43.56 |
| G | 80.02 | 56.44 |
| G/H | 12.33 | 1.30 |
| Structural moieties (%) | | |
| Ph-C1 | 27.19 | 34.24 |
| Ph-C2 | 20.80 | 14.77 |
| Ph-C3 | 6.28 | 2.35 |
| Ph-C1-2/ph-C3 | 7.64 | 20.84 |

The sum of relative abundance and structural classification are according to Table S2.

The Fenton-like reaction breakdown linkages between lignin subunits are also confirmed by phenolic compounds with various side chains obtained by Py-GC/MS. Compared to Kraft lignin, the ph-C1-2/ph-C3 ratio increased in the treated lignin, and the increase was mainly caused by the decrease of ph-C3 products (Table 1). In this study, the fast pyrolysis process (1 min) of lignin mainly yields aromatic compounds with different chain lengths (ph-C0-3) via bond cleavage because the secondary reactions are absent [30]. Therefore, the chain length of pyrolysis aromatic compounds is equal to the chain length of lignin subunits in their original form. Hence, the ph-C3 products can only be obtained from the lignin subunits with three carbon side chains. In the native lignin, β-O-4 linkages consisting of $C_\alpha$-$C_\beta$-$C_\gamma$ side chains are the most abundant linkage [31]; as a result, most of the ph-C3 products were obtained from the cleavage of β-O-4 linkages during the fast pyrolysis. These results proved the indirect evidence of β-O-4 linkages cleavage of lignin during the Fenton-like reaction treatment.

Second, in order to obtain more detailed structural information of Kraft lignin and Fenton-like-reaction-treated lignin, 2D NMR was conducted to investigate the changes of inter-unit linkage on lignin. The aliphatic ($\delta_C/\delta_H$ 50–90/2.5–6.0) region and corelated

assigned substructures of the lignin inter-unit linkages information are presented in Figure 3. The chemical shift assignments of $^1$H-$^{13}$C correlations of lignin are shown in Table S3. The β-O-4′ linkage (a) in Kraft lignin was 1.23 and decreased to 0.64 after the Fenton-like reaction treatment. Meanwhile, Kraft lignin had a content of β-5′ (phenylcoumarans, b) at 0.26 and β-β′ (resinols, c) at 0.53; however, no β-5′ and β-β′ linkages were detected in treated lignin by 2D NMR (Table 2). A high content of condensed C-C (β-5′ and β-β′) linked structure in Kraft lignin inhibited its bioconversion rate, which consisted of the low yield of PHA mentioned above (Figure 2A). The analytic results show that the relative abundances of C-O (β-O-4′) and C-C linkages in the treated lignin decreased compared to the Kraft lignin (Table 2). As shown in Table 1, less content of ph-C3 products was detected in the treated lignin, revealing the β-O-4 linkage breakdown during the treatment. These results together indicate that the bond cleavage occurred during the Fenton-like reaction treatment. The bond cleavage could result in the enhanced production of lignin-derived aromatic compounds and an improved microbial conversion yield of treated lignin [32]. The bond cleavage further showed that the cleavage of the recalcitrant C-C bonds was faster than the cleavage of weaker C-O bonds during the Fenton-like reaction treatment, leading to the decreased condensation degree, which was 0.43 in Kraft lignin and cannot be detected due to the absence of β-β′ linkage in the treated lignin (Table 2). The Fenton-like reaction system breaks down more C-C bonds and makes the inter-unit linkages in the residual lignin less compact, which results in less recalcitrance of lignin to microbial conversion. The methoxy (OMe) content in the Kraft lignin and treated lignin was 21.60 and 14.10% (Table 2), respectively. These data are consistent with the decreased G-type of lignin measured for treated lignin (Table 1), further proving that the demethylation or demothoxylation happened during the Fenton-like reaction system.

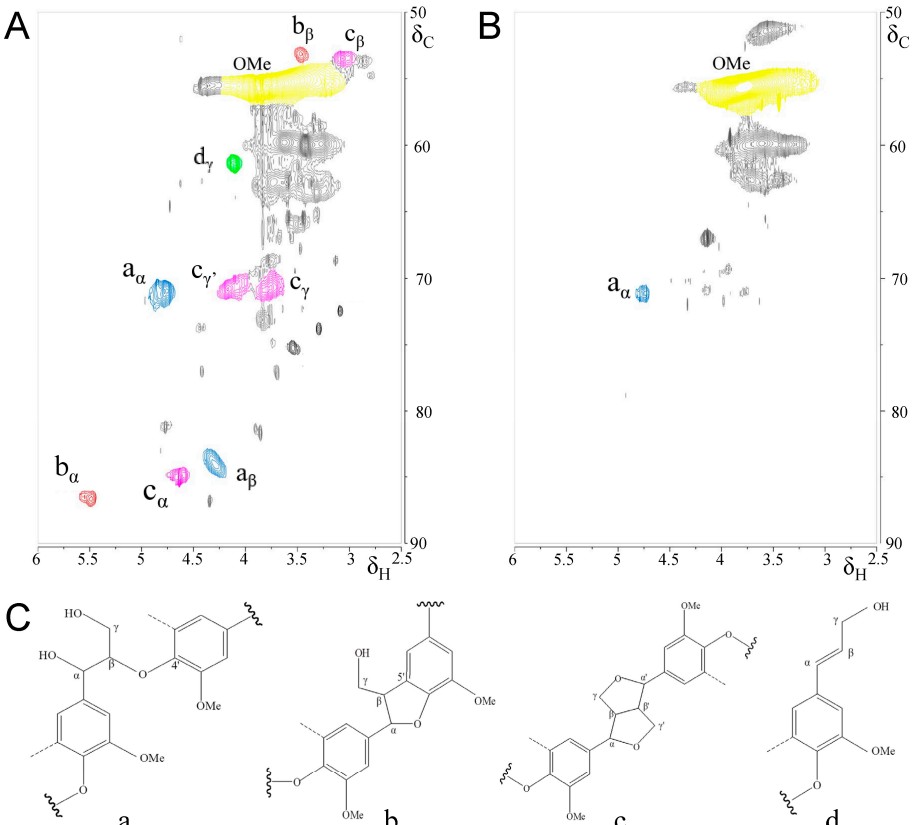

**Figure 3.** Aliphatic ($\delta_C$/$\delta_H$ 50–90/2.5–6.0) regions on 2D NMR spectra of (**A**) Kraft lignin and (**B**) treated lignin. (**C**) Lignin inter-unit linkages and end-group: (**a**) β-O-4′linkage, (**b**) phenyl-coumaran, (**c**) Resinol, and (**d**) cinnamyl alcohol end-group.

**Table 2.** Structural characterization in HSQC spectra and methoxy content of lignin and Fenton-like-reaction-treated lignin.

|  | Kraft Lignin | Treated Lignin |
|---|---|---|
| Lignin inter-unit linkages [1] |  |  |
| β-O-4′ | 1.23 | 0.64 |
| β-5′ | 0.26 | ND |
| β-β′ | 0.53 | ND |
| Lignin end-group [2] | 0.60 | ND |
| Condensation degree [3] | 0.43 | ND |
| OMe (%) | 21.62 | 14.10 |

ND not detected. [1] Linkages are relative to the methoxy. [2] Cinnamyl alcohol. [3] Ratio of (β-β′)/β-O-4′.

Third, GC-MS analysis was used to investigate the lignin depolymerized products (mono-aromatic and short-chain organic compounds) from the Fenton-like reaction system (Figure 4). The depolymerized products were enriched in acid, alcohol, and aldehyde. As shown in Figure 4, the acid products, phenyl propane (ph-C3) compounds (p-coumaric acid and ferulic acid), phenyl methane (ph-C1) compounds (3,4-dihydroxybenzoic acid and vanillic acid), and short-chain organic compounds (such as lactic acid, succinic acid, etc.) were significantly increased after the Fenton-like reaction. For example, the content of p-hydroxybenzoic acid, vanillic acid, and lactic acid was 0.1, 2.8, and 1.2% on Kraft lignin and 9.8, 14.4, and 2.7% on treated lignin, respectively, which increased 75-, 4-, and 1-fold after the Fenton-like reaction treatment. The alcohol and aldehyde products significantly decreased from 27.8 to 12.5% and from 13.1 to 7.4%, respectively. Meanwhile, the vanillin content in the treated lignin was reduced compared to the Kraft lignin, whereas the vanillic acid content was increased after treatment. The oxidation of vanillin to generate vanillic acid has been reported previously for oxidative treatments of lignin [33]. These results confirm that the Fenton-like reaction treatment resulted in the formation of mono-aromatic and short-chain organic acid compounds through bond cleavage.

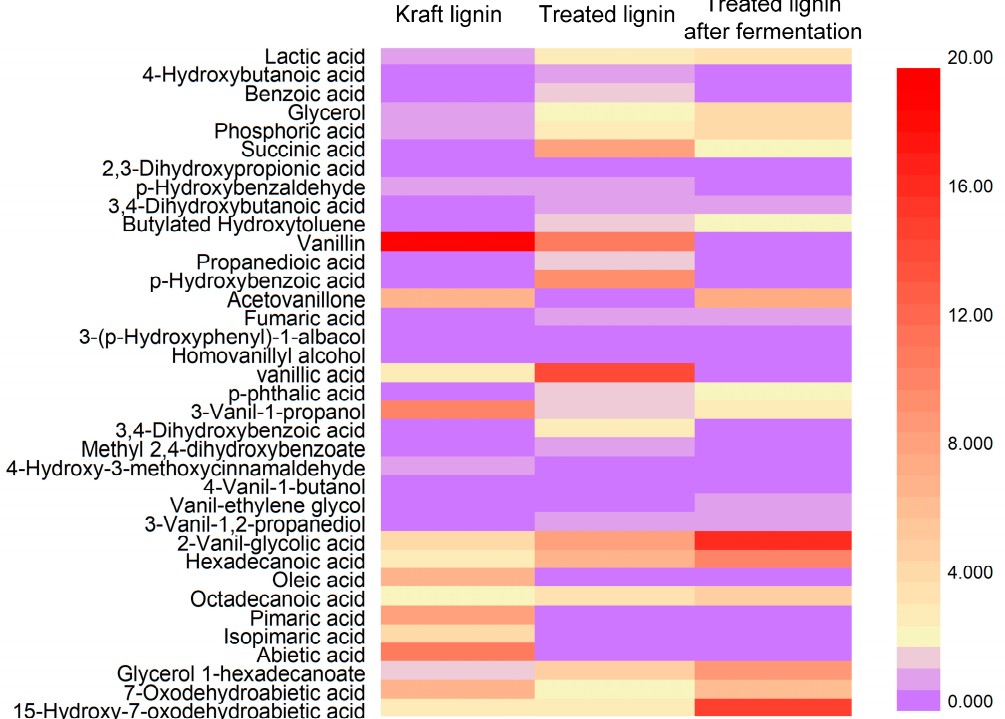

**Figure 4.** Variation of soluble compounds of Kraft lignin, treated lignin, and residue of treated lignin after microbial fermentation that detected by GC/MS.

### 3.4. Growth of P. putida KT2440 and PHA Accumulation on Lignin Depolymerized Products

As shown in Figure 4, various lignin depolymerized products were decreased or even could not be detected after microbial conversion, including the phenyl propane compounds, phenyl methane compounds, and short-chain organic compounds, which were significantly decreased. Based on this, seven of the products (p-coumaric acid, ferulic acid, p-hydroxybenzoic acid, vanillic acid, succinic acid, lactic acid, and glycerol) from the lignin depolymerization and glucose were chosen for their toxicity and the bioconversion test.

As shown in Figure 5A, these lignin depolymerized products could be efficiently utilized by *P. putida* KT2440 because less than 10% of products were left after 72-h fermentation. Meanwhile, we found *P. putida* KT2440 could be well-grown in the medium using glucose and all the lignin depolymerized products as the solo carbon source (Figure 5B). Among the mono-aromatic compounds, *p*-coumaric acid and *p*-hydroxybenzoic acid with less methoxy group were preferred to be utilized due to their higher bacterial biomass than ferulic acid and vanillic acid, respectively. The bacterial biomass obtained from *p*-coumaric acid and *p*-hydroxybenzoic acid was 672.0 ± 11.0 (Figure 5(B1)) and 627.3 ± 21.0 mg/L (Figure 5(B2)), which was comparable to glucose (683.0 ± 13.1 mg/L, Figure 5(B1)). For short-chain organic compounds, lactic acid and glycerol showed more cell growth than succinic acid (Figure 5(B3)). As previously reported, aldehydes and alcohols from lignin depolymerization showed the highest toxicity to microorganisms, whereas carboxylic acids were the lowest [34]. *P. putida* KT2440 showed considerable growth on these lignin depolymerized acid products, indicating that the major depolymerized compounds from the Fenton-like treatment imposes low or no toxicity to *P. putida* KT2440.

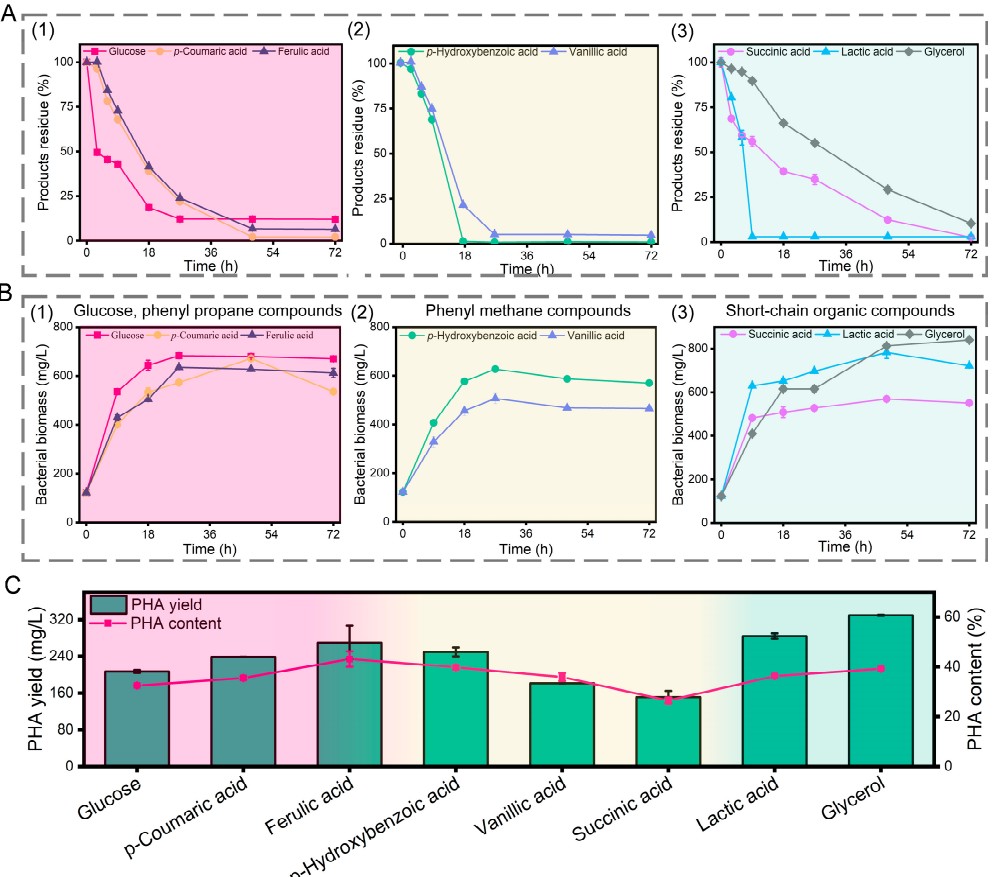

**Figure 5.** The toxicity and bioconversion of lignin depolymerized products. (**A**) Products residue and (**B**) bacterial biomass after different fermentation time. (**C**) The highest PHA yield and PHA content obtained from eight types of products fermented by *P. putida* KT2440.

As mentioned above, the Fenton-like reaction treatment can efficiently improve the PHA accumulation from the treated lignin, which might be due to the reduced toxicity of lignin-depolymerized products on cell growth and PHA production. The major products from lignin depolymerization have been proved to have low toxicity to bacterial growth, and their ability for PHA accumulation was further tested. The PHA accumulation of glucose and seven lignin depolymerized products increased with fermentation time. The maximum PHA yield on phenyl methane compounds was at 27 h; for the other compounds, after 48 h of fermentation, the PHA reached the highest content (Figure S1). As shown in Figure 5C, *P. putida* KT2440 had a PHA yield and PHA content of 207.1 ± 3.2 and 32.5 ± 0.2, 238.7 ± 0.5 and 35.5 ± 0.7, 270.1 ± 36.6 and 43.1 ± 6.1, 249.5 ± 9.6 and 39.8 ± 0.8, 181.4 ± 0.4 and 35.9 ± 1.6, 151.2 ± 13.6 and 26.5 ± 1.2, 284.1 ± 6.0 and 36.3 ± 0.5, 329.6 ± 1.2 mg/L and 39.2 ± 0.0% on glucose, *p*-coumaric acid, ferulic acid, *p*-hydroxybenzoic acid, vanillic acid, succinic acid, lactic acid, and glycerol, respectively. Except for vanillic acid and succinic acid, the PHA yield obtained from the other five kinds of lignin-depolymerized products were 1.14–1.58-fold higher than that of glucose. Compared to phenyl methane compounds, the phenyl propane compounds led to higher PHA yield after fermentation, which was consistent with a previous report that *p*-coumaric acid had a higher PHA production than 4-hydroxybenzoic acid in *Burkholderia* sp. ISTR5 fermentation [35]. These results together indicate that *P. putida* KT2440 can utilize most of the lignin-depolymerized products treated from the Fenton-like reaction as a solo carbon source to grow and produce PHA, and the PHA yield varied with the product species.

Based on the above analysis of lignin structures, lignin-depolymerized products, and microbial conversion of lignin, we summarized that the Fenton-like reaction treatment provides more available and fermentable carbon sources for *P. putida* KT2440 by the bond cleavage of lignin and the formation of mono-aromatic and short-chain organic acid compounds, further improving the microbial conversion of treated lignin into PHA.

*3.5. Fenton-like Reaction Treatment of Black Liquor for PHA Production*

The Fenton-like reaction has proved its capacity to depolymerize Kraft lignin for improving microbial conversion efficiency, and we further demonstrated its potential for treating black liquor directly from the pulping industry. As shown in Figure 6A, the pH of black liquor was adjusted to 7, and then the black liquor divided into two phases, the liquid solution and the solid residue. The soluble lignin, mono-aromatic, and low molecular compounds were still in the liquid phase and condensed lignin was precipitated in the sediment. The composition of the compound in the liquid phase is shown in Table S4 and the soluble lignin structure is shown in Figure S2 and Table S5. The liquid phase had a high content of mono-aromatic (15.1%) and small organic (84.0%) compounds; the content of most of these compounds decreased or even could not be detected after 27-h fermentation (Table S4), suggesting that these consumed compounds were used to support the PHA accumulation. As a result, a high yield of PHA (770.5 mg) was obtained from the liquid phase, which we obtained from the 1-L black liquor (Figure 6B). The soluble lignin in black liquor had a high condensation degree, and the linkages in the lignin were slightly decreased after the fermentation (Table S5), indicating that *P. putida* KT2440 can barely convert lignin in the black liquor. Therefore, the Fenton-like reaction was hired to depolymerize the condensed lignin in the solid residue. As shown in Figure 6B, the PHA yield of treated residue was significantly higher than that of solid residue, in which 76.7 mg of PHA was obtained from the solid residue; however, 149.9 mg of PHA was obtained from the treated residue and increased by 98.4% compared to the solid residue. Overall, the total PHA yield of 1-L black liquor was 920.4 mg with the Fenton-like reaction treatment. This result of black liquor microbial conversion highlights the potential of Fenton-like-reaction-assisted lignin depolymerization for practical application in PHA production.

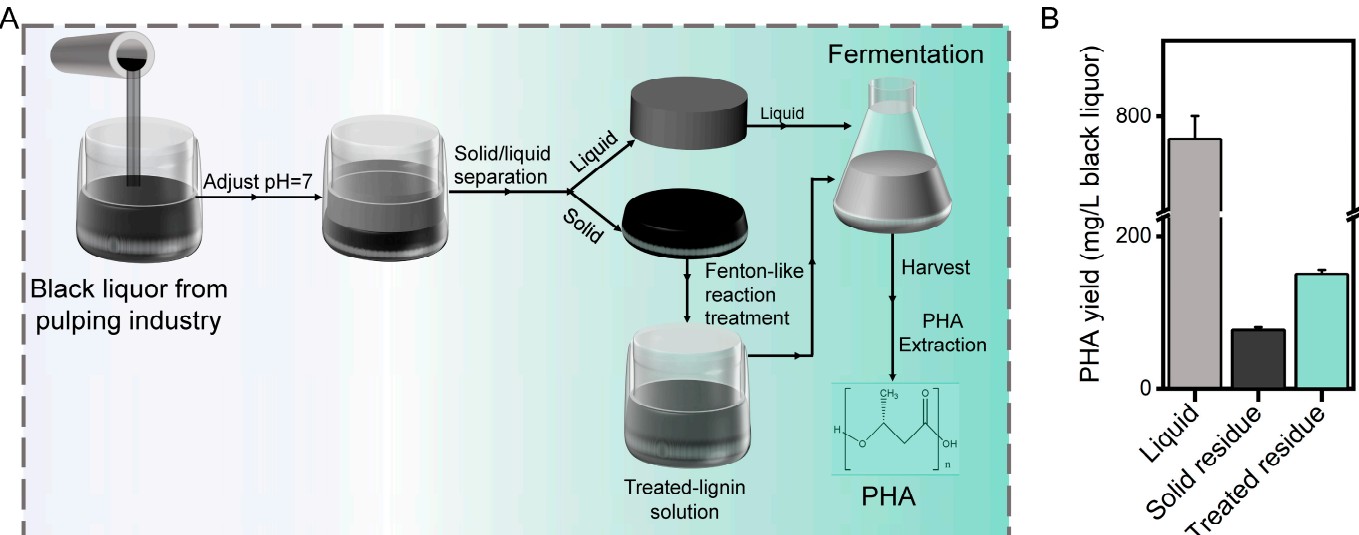

**Figure 6.** Black liquor bioconversion by P. putida KT2440. (**A**) Schematic of Fenton-like reaction assisted black liquor fermentation. (**B**) The PHA yield obtained from different phases of black liquor after microbial conversion.

## 4. Conclusions

This study firstly combines the advantage of fungal oxidation and bacterial conversion of lignin for PHA production. The Fenton-like reaction mimicking the radical-mediated oxidation process of fungi was found to depolymerize lignin at natural pH. The biomimetic treatment can depolymerize Kraft lignin into mono-aromatic and short-chain organic compounds through bond cleavage, which provide less toxic and more available carbon sources for bacteria to utilize. The PHA yield of *P. putida* KT2440 fermented from either Kraft lignin or black liquor was greatly improved by the Fenton-like reaction, which demonstrates the potential of this biomimetic oxidative reaction for practical applications. Overall, this study indicated that biomimetic treatment based on wood decay fungi and ligninolytic bacteria can synergize to efficiently convert lignin by promoting the depolymerization of recalcitrant Kraft lignin, thus providing a novel strategy for efficient lignin valorization. Future studies could employ synthetic biological methods to improve the biotransformation capabilities of this bioconversion process and further integrate with advanced nanotechnologies to upgrade lignin fermentation residues into multifunctional nanomaterials to completely valorize the lignin into high-value products, as we discussed in our published review papers [36,37].

**Supplementary Materials:** The following supporting information can be downloaded at: https://www.mdpi.com/article/10.3390/fermentation9070649/s1, Figure S1: PHA yield and PHA content of glucose and lignin-depolymerized products; Figure S2: Aliphatic ($\delta C/\delta H$ 50–90/2.5–6.0) regions on 2D HSQC NMR spectra of (A) soluble lignin in the black liquor and (B) lignin residue after fermentation; Table S1: The concentration of reagent under different reaction conditions; Table S2: Relative abundance of the aromatic-compounds-derived peaks identified in the Py-GC/MS of Kraft lignin and Fenton-like-reaction-treated lignin; Table S3: The assignments of $^1$H-$^{13}$C peaks in HSQC spectra; Table S4: GC/MS analysis of soluble compounds in liquid phase of black liquor before and after the *P. putida* KT2440 fermentation; Table S5: Structural characterization in HSQC spectra and methoxy content of soluble lignin in the black liquor.

**Author Contributions:** Conceptualization, S.X., H.Y., X.F. and Q.G.; data curation, X.F., Q.G., X.L. and Z.Z.; formal analysis, X.F.; funding acquisition, X.Z.; investigation, X.F.; methodology, Q.G.; project administration, H.Y.; resources, X.F.; software, X.F.; supervision, F.M., S.X. and H.Y.; validation, X.F., Q.G. and X.L.; writing—original draft, Q.G.; writing—review and editing, X.F. All authors have read and agreed to the published version of the manuscript.

**Funding:** This research was funded by the National Key R&D Program of China (2019YFA0905504).

**Institutional Review Board Statement:** Not applicable.

**Informed Consent Statement:** Not applicable.

**Data Availability Statement:** All data will be made available upon request.

**Acknowledgments:** The authors thank the Centre of Analysis and Test of Huazhong University of Science for Py-GC/MS and GC/MS analysis.

**Conflicts of Interest:** The authors declare no conflict of interest.

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
