# Peer review of "Mimicking the Fungal Decay Strategy for Promoting the Bacterial Production of Polyhydroxyalkanoate from Kraft Lignin"

_fermentation, doi:10.3390/fermentation9070649_

Round 1

Reviewer 1 Report

Minor corrections

L106: 0.2 mL

L111: concentration

L114: 20 mL

L132: polytetrafluorethylene

Author Response

We appreciated very much the recognition from the reviewer that the manuscript needs minor corrections. As shown below, we have addressed the comments one-by-one, and we believe all these constructive comments and revisions have helped us to significantly improve our manuscript.

Point 1: L106: 0.2 mL

Response 1: We appreciated the comment very much. As suggested, we have corrected the word 0.2 ml’ into ‘0.2 mL’ in Page 3 Line 106. And we have checked the entire manuscript and corrected the similar mistake. In Page 4 Line 149, we have changed the word 0.75 ml’ into 0.75 mL’.

Point 2: L111: concentration

Response 2: We appreciated the comment very much. As suggested, we have changed the word in Page 3 Line 111 to concentration’.

Point 3: L114: 20 mL

Response 3: We appreciated the comment very much. As suggested, we have changed the word 20 ml’ into ‘20 mL’ in Page 3 Line 114.

Point 4: L132: polytetrafluorethylene

Response 4: We appreciated the comment. As suggested, we have deleted the letter ‘μ’ in Page 3 Line 132, and this word has been changed into polytetrafluorethylene’.

Reviewer 2 Report

The biological conversion of lignin is an interesting topic because it is rarely reported compared to chemical/thermal methods. The approach of integrated fungal oxidation and bacterial conversion for PHA production is interesting. The manuscript is well-written overall and is expected to engage the reader's attention.

1. How much energy consumption is expected to be saved by this developed process compared to existing chemical or thermal pretreatment methods?

2. What are the considerations and avenues for future research to reduce microbial inhibitors and enhance biotransformation?

3. How can residues be utilized after bioconversion for the whole conversion?

Author Response

The biological conversion of lignin is an interesting topic because it is rarely reported compared to chemical/thermal methods. The approach of integrated fungal oxidation and bacterial conversion for PHA production is interesting. The manuscript is well-written overall and is expected to engage the reader's attention.

Response: We appreciated very much the recognition from the reviewer that ‘the approach of integrated fungal oxidation and bacterial conversion for PHA production is interesting’ and ‘the manuscript is well-written and is expected to engage the reader’s attention’. As shown below, we have carefully addressed each comment from the reviewer one-by-one.

Point 1: How much energy consumption is expected to be saved by this developed process compared to existing chemical or thermal pretreatment methods?

Response 1: We appreciated the comment very much. It is very difficult to provide an exact estimate of the savings compared to existing chemical or thermal pretreatment methods for lignin valorizatiaon in current stage. The energy savings can vary depending on the specific design and efficiency of the developed process. Our study using Fenton-like reaction operate under milder conditions, requiring lower temperature under ambient pressure; the chemical or thermal pretreatment need higher temperature or pressures which are the energy-intensive preprocessing steps, as a result, the Fenton-like reaction can save energy by reducing the heating and pressurization process. Of course, further condition and processing optimization should be conducted to achieve for a well-developed economical PHA production from lignin.

Point 2: What are the considerations and avenues for future research to reduce microbial inhibitors and enhance biotransformation?

Response 2: We appreciated the comment very much. We think the high-throughput screening approaches combined with directed evolution can be employed to rapidly identify microbial strains with enhanced resistance to inhibitors; then explore potential resistance genes through differential analysis of genomics and transcriptomics; finally decipher the key mechanism for microbial resistance of inhibitors. Additionally, the synthetic biological methods are powerful tools for reducing microbial inhibitors and enhancing biotransformation [1]. Currently, we are working on exploring and modifying microbial strains to enhance their tolerance to inhibitors or improve their biotransformation capabilities. For instance, we are employing gene editing tools such as CRISPR-Cas9 and regulatory elements like promoters and regulators to precisely modify the genome of Pseudomonas putida KT2440. By enhancing its relevant metabolic pathways, we aim to expand its ability to utilize various types of lignin valorization products and improve its biotransformation capabilities. As suggested, we have added the sentence to discuss the future research to enhance biotransformation, which is in Page 12 Line 452-453 and reads as below:

‘Future studies could employ the synthetic biological methods to improve the biotransformation capabilities of this bioconversion process’.

Ref:

[1] Li, F., Zhao, Y., Xue, L., Ma, F., Dai, S. Y., Xie, S. Microbial lignin valorization through depolymerization to aromatics conversion. Trends Biotechnol. 2022, 40, 1469-1487.

Point 3: How can residues be utilized after bioconversion for the whole conversion?

Response 3: We appreciated the comment. We think it is a great idea to convert lignin residues after bioconversion process into agricultural products to enhance the soil carbon sequestration [2]. To use lignin waste by integrating advanced nanotechnologies and agricultural biotechnologies to upgrade it into multifunctional nanomaterials for agriculture practicing, including nano-carriers for herbicides, pesticides, and fertilizer, nano-biochars for plant growth, plant protection, crop production, and soil health, and nano-composite as a biodegradable alternative of traditional plastic agricultural mulching. This strategy can not only convert lignin waste into valuable products but also enhance carbon sequestration on sustainable agriculture to alleviate the global food crisis caused by soil carbon loss. According, we have added the sentence to discuss the future work for residue lignin conversion, which is in Page 12 Line 452-456 and reads as below:

‘Future studies could employ the synthetic biological methods to improve the biotransformation capabilities of this bioconversion process and further integrate with advanced nanotechnologies to upgrade lignin fermentation residues into multifunctional nanomaterials to completely valorizating the lignin to high-value products as we discussed in our published review papers’.

Ref:

[2] Fu, X., Zheng, Z., Sha, Z., Cao, H., Yuan, Q., Yu, H., Li, Q. Biorefining waste into nano-biotechnologies can revolutionize sustainable agriculture. Trends Biotechnol. 2022, 40, 1503-1518.